# Domain-Generalizable Multiple-Domain Clustering

**Amit Rozner**[*]
*Faculty of Engineering, Bar Ilan University*
[*] *Equal contribution*

**Barak Battash**[*]
*Faculty of Engineering, Bar Ilan University*
[*] *Equal contribution*

**Lior Wolf**
*School of Computer Science, Tel Aviv University*

**Ofir Lindenbaum**                                                    *ofirlin@gmail.com*
*Faculty of Engineering, Bar Ilan University*

**Reviewed on OpenReview:** *https: // https: // openreview. net/ forum? id= O9RUANpPmb*

## Abstract

This work generalizes the problem of unsupervised domain generalization to the case in which no labeled samples are available (completely unsupervised). We are given unlabeled samples from multiple source domains, and we aim to learn a shared predictor that assigns examples to semantically related clusters. Evaluation is done by predicting cluster assignments in previously unseen domains. Towards this goal, we propose a two-stage training framework: (1) self-supervised pre-training for extracting domain invariant semantic features. (2) multi-head cluster prediction with pseudo labels, which rely on both the feature space and cluster head prediction, further leveraging a novel prediction-based label smoothing scheme. We demonstrate empirically that our model is more accurate than baselines that require fine-tuning using samples from the target domain or some level of supervision. Our code is available at `https://github.com/AmitRozner/domain-generalizable-multiple-domain-clustering`.

## 1 Introduction

Clustering high-dimensional measurements accurately is crucial for effectively analyzing scientific data. In recent years, deep learning-based frameworks have shown significant improvements when compared to classical clustering models, such as $K$-means (Lloyd, 1982) or Spectral Clustering (Ng et al., 2001). However, if the observations are collected from multiple domains, classical and deep learning clustering models may group samples based on domain-specific information, such as style, rather than their semantic content. This problem becomes even more difficult if samples from the target domain are not available during training. This situation may arise due to privacy or

computational considerations and has several applications, such as face clustering on edge devices (Caldarola et al., 2021), medical image clustering (Irshaid et al., 2022), and more.

To overcome this challenge, we develop a predictor called $f$ that can categorize a given sample to its respective cluster, regardless of its domain. We then evaluate $f$ on an unseen target domain where no samples were seen during training. This task requires solving the problem of multiple-domain clustering along with domain generalization. The combination of these two problems is highly valuable, especially in scientific discovery in fields such as biomedicine, where individuals have different distributions (Zhao et al., 2021). Accurately predicting new domains is highly valuable in these fields (Bae et al., 2022; Farhadian et al., 2022). While previous studies have tackled similar problems, they rely on partial supervision or require access to target domain samples during training (Zhang et al., 2021; Harary et al., 2022; Gopalan, 2017; Menapace et al., 2020).

Formally, we are given a dataset $S$ with $N$ samples from $d$ domains, where each domain has $N_i, \forall i \in \{1, ..., d\}$ samples. We know which sample belongs to each domain, and a classifier $f$ is trained to map every sample in $S$ to one of $K$ groups. For evaluation purposes, a set of labeled examples is used, and the $K$ groups are assigned to a set of ground truth labels using a best-matching method, the Hungarian method (Kuhn, 1955), on an unseen single-domain dataset $T$. To clarify, without further training, $f$ is applied to each sample of $T$, and the predicted labels are then compared to the ground truth labels of the samples in $T$.

We propose a two-phase learning approach to tackle the challenge of clustering data from multiple domains while ensuring domain generalization. In the first phase, called the *pretraining stage*, we use a contrastive loss with several complementary elements to encourage learning domain-agnostic representations. We achieve this by using style augmentations that reduce stylistic variations and an adversarial domain loss that eliminates the influence of domain-specific content. We also introduce a special queue procedure and domain balancing to stabilize the multi-domain training scheme. In the second phase, we present several novel components to learn multi-domain *clustering assignments*. These include (i) a new pseudo-label selection procedure that uses a style-augmented common domain to identify reliable labels, (ii) a prediction-based label smoothing procedure that prevents mode collapse to highly populated clusters and can overcome bad initialization, and (iii) a multiple clustering head prediction and filtration procedure to stabilize the clustering step in the presence of high-dimensional data or many categories.

We have tested our method on various multi-domain datasets and have demonstrated its superiority over several strong baselines. Our framework is comparable to or better than state-of-the-art clustering methods but does not require any labeled data or adaptation to samples from the target domain. By removing these two requirements, our model opens the door to new applications where labeled data is unavailable or access to target samples is infeasible due to privacy constraints. Furthermore, our ablation studies confirm the importance of each proposed component in contributing to the model's generalization capabilities.

## 2 Background and related work

**Self-supervised learning (SSL)** is a technique for extracting useful representation from unlabelled data using a pretext task. In computer vision, many methods rely on a contrastive loss; for example, SimCLR (Chen et al., 2020a) uses paired augmentations, creating feature representations that are projected and trained to maximize agreement. The seminal SSL framework MoCo (He

et al., 2020) introduced a queue of negative samples and a momentum-moving average encoder to improve the contrastive learning framework.

**Deep clustering** aims to exploit the strength of neural networks (NN) as feature extractors to identify a representation that better preserves cluster structures using unlabelled data. DeepCluster (Caron et al., 2018) is a method that learns useful features and assigns them to clusters by applying $k$-means to the extracted features and training a NN to predict the cluster assignments. Ji et al. (2018) use content-preserving image transformations to create pairs of samples with shared semantic information. Then, they train a NN to maximize the mutual information between the image pairs in a probabilistic data representation. Recently, Semantic Pseudo-Labeling for Image Clustering (Niu et al., 2022) has obtained state-of-the-art results on several benchmarks. This method is an iterative deep clustering approach that relies on self-supervision and pseudo-labeling. First, self-supervision is performed using a contrastive loss to learn informative features. Then, prototype pseudo-labeling is created to avoid common miss-annotations in pseudo-labeling techniques.

**Domain generalization (DG)** is a technique that involves incorporating knowledge from multiple labeled source domains into a model that can perform well on unseen target domains. Cha et al. (2021) have developed a dense and overfit-aware stochastic weight sampling strategy, which seeks a flat minimum and has been found to reduce the domain gap and prevent overfitting. Zhou et al. (2021) have proposed a method that aims to mix the styles of different image domains by using a probabilistic convex combination between instance-level feature statistics of early CNN layers. Li et al. (2021) uses augmentations to mitigate the domain gap by perturbing the feature embedding with Gaussian noise during training. Zeng et al. (2023) have focused on "evolving domain generalization" with additional information of the time, which helps with the class label prediction in the target domain.

**Unsupervised domain generalization (UDG)** was recently presented by Zhang et al. (2021); Harary et al. (2022). UDG is related to our goal but requires some supervision. Specifically, UDG involves: unsupervised training on a set of source domains, then fitting a classifier using a small number of labeled images. Ultimately, the model is evaluated on a set of target domains that were unseen during training. Toward this goal, Zhang et al. (2021) suggested ignoring domain-related features by selecting negative samples from a queue based on their similarity to the positive sample domain. Harary et al. (2022) recently presented BrAD, which involves self-supervised pre-training on multiple source domains in a shared sketch-like domain. To fine-tune a classifier, they used varying amounts of labeled samples from source domains. In contrast, our research does not use class labels for either source or target domains during model training. Additionally, we assume no knowledge about corresponding images between domains but only that all target classes appear in each domain.

**Unsupervised Clustering under Domain Shift (UCDS)** presented by Menapace et al. (2020) discusses a task of clustering samples from multiple source domains and adapting the model to the target domain using multiple unlabeled samples from that domain. The authors propose optimizing an information-theoretic loss, along with domain-alignment layers, to achieve this goal. Figure 1 shows different approaches for multi-domain image classification. Although the UCDS setting is similar to ours, we aim to design a model that can predict cluster assignments on multiple source domains and generalize to new unseen domains without any further tuning or adaptation. This is the first work that solves this task without using any labels or samples from the target domain, which is

a significant advantage in real-world settings where we often don't know that a domain shift occurred or can't access samples from the test domain due to privacy or computational considerations.

| Source Domains | | Target Domain | Training Data Required | | |
|---|---|---|---|---|---|

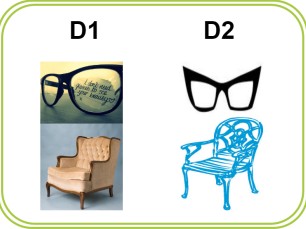 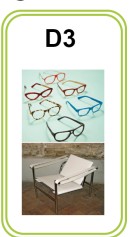

| Task | Source | Target |
|---|---|---|
| **DA** | LABELLED | LABELLED |
| **UDA** | LABELLED | UNLABELLED |
| **DG** | LABELLED | - |
| **UDG** | LIMITED | - |
| **UCDS** | UNLABELLED | UNLABELLED |
| **Ours** | UNLABELLED | - |

Figure 1: Different tasks for multi-domain image classification or clustering. Domain adaptation (DA) deals with adapting a model from one domain to another and requires the availability of labeled samples in the source and target domain labels for training the model (Csurka et al., 2017). Unsupervised domain adaptation (UDA) alleviates the need for labeled target samples and is trained using labeled source samples and unlabeled samples from the target domain (Piva et al., 2023). Domain generalization (DG) aims to create a model that generalizes across multiple source and unseen target domains. The training procedure is similar to UDA but does not require access to samples (unlabeled) from the target domain during training (Roy et al., 2019; Matsuura & Harada, 2020). Unsupervised domain generalization (UDG) does not assume access to target samples but relies only on a limited amount of labeled data in the source domain (Harary et al., 2022; Zhang et al., 2022). Unsupervised clustering under domain shift (UCDS) does not rely on labeled data but requires access to samples from the source and target domains for training (Menapace et al., 2020). Our fully unsupervised approach can generalize to the target domain without access to its samples during training.

## 3 Method

Our approach involves two phases. In the first phase, we train a feature extraction model, denoted as $f_e$, in a self-supervised manner on data from various source domains. This phase helps to bridge the gap between different domains by extracting semantically related features $u_i \in \mathbb{R}^e$, where $e$ represents the embedding dimension. In the second phase, we focus on training a clustering head, denoted as $f_c$, while the weights of the feature extraction model $f_e$ are kept frozen. Our cluster predictions are then based on the composition of $f_e$ and $f_c$, denoted as $f = f_c \circ f_e$.

The feature extraction process has several pre-training stages that aim to learn features that capture semantic information from each image while suppressing the style content. In order to train a cluster predictor that is invariant to the domain, we suggest the following components: data augmentation for training the clustering head, generating reliable pseudo-labels, using a clustering head training loss, smoothing based on predictions, and using multiple clustering heads. A detailed explanation of each of these steps can be found in the following subsections.

### 3.1 Pre-training

Our pre-training strategy generalizes MoCoV2 (Chen et al., 2020b) for multiple domains by introducing four components that enable learning robust domain-agnostic embeddings. First, we introduce style transfer augmentations, and using a contrastive loss, we learn a representation invariant to stylistic variations across domains. We further introduce an adversarial loss to eliminate domain-specific information at the distribution level. To stabilize the multi-domain contrastive training process, we introduce domain-specific queues, each tailored to a distinct domain. This prevents pushing apart representations from different domains and encourages learning content rather than domain-specific features. Additionally, we address domain imbalance by implementing a virtual balancing approach to mitigate the impact of population differences across domains. These pre-training components are detailed below, followed by a description of the contrastive learning framework used here.

1. *Style transfer augmentation:* We propose incorporating style augmentations into our contrastive loss to attenuate the influence of "style" on our extracted representation. Specifically, with probability $p_{st}$ we perform a style transfer augmentation $\mathcal{ST}(x_i)$. This example is then fed into a contrastive loss with $x_i$ that learns a mapping such that $\mathcal{ST}(x_i)$ and $x_i$ are indistinguishable. To augment with varying styles of domains, we use Huang & Belongie (2017); this allows us to enhance the ability of our feature extractor to ignore domain-specific features.

2. *Adversarial loss:* We add a complementary step to remove domain-specific information at the distribution level. We achieve this using an adversarial domain loss (Ganin & Lempitsky, 2015). Specifically, we train our feature extractor $f_e$ to learn features that are not influenced by domain identity. To do this, we "fool" a domain classifier $f_d$ by updating the weights of $f_e$ using a constant $\lambda_d \geq 0$ and the gradient of $f_d$ denoted as $\frac{\partial \mathcal{L}_{f_d}}{\partial \theta_d}$. This technique is called a gradient reversal layer. This encourages the feature extractor to remove the domain-specific information. We note that the label of the domain is generally available for free; for instance, in medical applications, this could refer to the imaging technology.

3. *Domain-specific queues:* In contrastive training, negative samples are stored in queues. To avoid pushing apart the representations of different domains, we use multiple domain-specific queues (Harary et al., 2022). Specifically, we introduce domain-specific queues $Q = [Q_1, Q_2, ..., Q_d]$ each with size $N_q$ for each of the $d$ domains. The negative examples $u_i^-$ are drawn from the same domain as the positive sample, which makes the discrimination more challenging and encourages the model to focus on the content rather than the domain. Positive samples $u_i^+$ are stored in the relevant domain queue for later use as negative examples.

4. *Domain balancing:*

The populations of samples from different domains may differ significantly. Such domain imbalance can cause poor generalization since the model might predominantly learn from the heavily populated domain. To address this issue, we employ a virtual balancing approach by sampling domain-balanced mini-batches. This strategy helps mitigate the domain imbalance, enhancing the model's ability to generalize more effectively across the different domains.

**Pretraining overview** We are given an input batch of images $x = [x_1, x_2, .., x_B]^T \in \mathbb{R}^{B \times C \times W \times H}$ where $C, W, H$ represent color, width and height dimensions. Each image $x_i$ is transformed twice. First, using a strong augmentation $x_i^s = \mathcal{S}(x_i)$. Second, the image is transformed

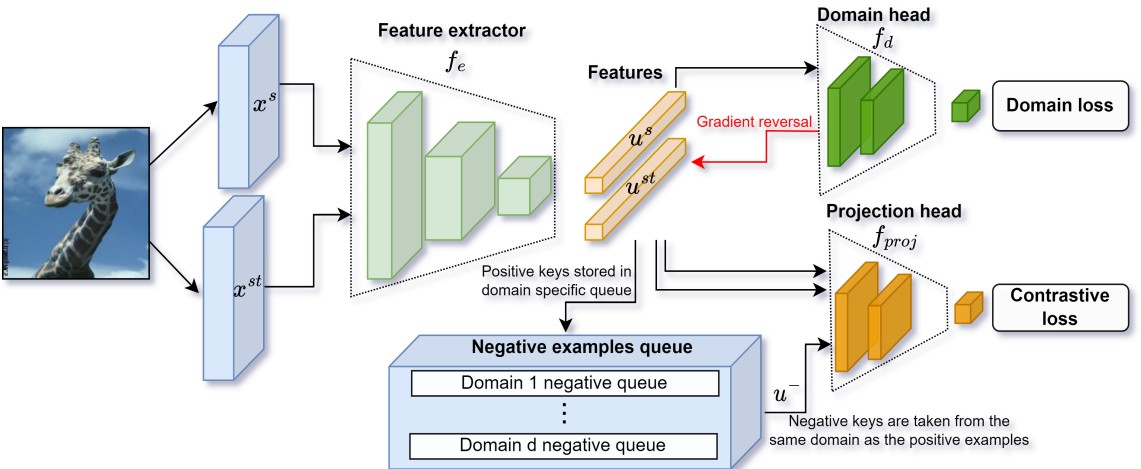

Figure 2: The proposed pre-training procedure. Each image is transformed using strong augmentations or style transfer augmentation. The features $u^s$ (strong augmentation) and $u^{st}$ (style augmentation) are extracted using $f_e$. Then we use a domain head $f_d$ to classify the domain identity of each sample, minimizing the domain loss $\mathcal{L}_{f_d}$; we use gradient reversal to update the feature extractor $f_e$ to fool the domain head in an adversarial fashion. The contrastive loss $\mathcal{L}_{f_{proj}}$ is minimized based on the output of the projection head $u^s$, $u^{st}$, and $u^-$ (negative samples).

using:

$$x_i^{st} = \begin{cases} \mathcal{ST}(x_i), & w.p \quad p_{st}, \\ \mathcal{S}(x_i) \quad , & w.p \quad 1 - p_{st}. \end{cases}$$

Where $\mathcal{ST}(x_i)$ replaces the style of $x_i$ with another domain's style with probability $p_{st}$, otherwise, a strong augmentation $\mathcal{S}(x_i)$ is applied to generate the positive sample. Both $x_i^s$ and $x_i^{st}$ are passed through the feature extractor $f_e$ to create the embeddings $u_i^s = \mathcal{P}(f_e(x_i^s, \theta_1), \theta_p)$, $u_i^{st} = \mathcal{P}(f_e(x_i^{st}, \theta_1'), \theta_p')$, where $\mathcal{P}$ and $\theta_p$ are the projection head and its weights respectively. $\theta_1' = \mu\theta_1' + (1-\mu)\theta_1$ and $\theta_p' = \mu\theta_p' + (1-\mu)\theta_p$ are the moving average versions of $\theta_1$ and $\theta_p$ respectively. Finally, negative samples $u_i^-, \forall i \in [0, N_q]$ are sampled from a domain queue of the same domain $d_{x_i}$ as $x_i$. We use the following contrastive loss:

$$\mathcal{L}_{f_{proj}} = -log \frac{exp((u^s)^T u^{st})}{\sum_{i=1}^{N_q} exp((u^s)^T u_i^-) + exp((u^s)^T u^{st})}, \tag{1}$$

where $u^s = [u_1^s, u_2^s, .., u_B^s]^T \in \mathbb{R}^{Bxe}$, $u^{st} = [u_1^{st}, u_2^{st}, .., u_B^{st}]^T \in \mathbb{R}^{Bxe}$, are the embeddings for the transformed input batch $x^s$, $x^{st}$.

To further remove domain-specific information, we use an additional domain adversarial term using a cross-entropy loss: $\mathcal{L}_{f_d} = -\mathcal{L}_{ce}(f_d(u^s), \theta_d)$. Hence, our complete objective is

$$\mathcal{L}_{f_e} = \mathcal{L}_{f_{proj}} + \mathcal{L}_{f_d}.$$

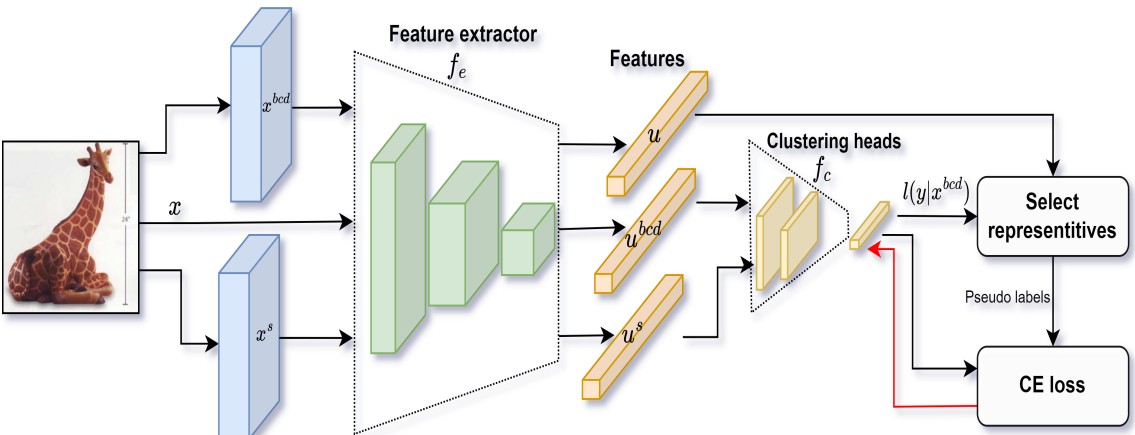

Figure 3: Clustering head training. The image is passed through the feature extractor $f_e$ in its original $(u)$, strongly augmented $(u^s)$, and BCD form $(u^{bcd})$. The weights of $f_e$ are frozen and used to produce the features. Representatives are selected from the original image features based on the clustering head's predictions over the BCD images. The class representatives are used as pseudo labels for the CE loss.

The contrastive loss term and domain-adversarial loss term help learn features that are related to the content and invariant to the domain.

## 3.2 Clustering predictor head

The pre-training phase provides a solid and robust feature extractor $f_e$ on which a clustering head $f_c$ can be trained. The clustering head should be able to separate multi-domain data into groups of samples with similar semantic content while ignoring the cross-domain distribution shifts. We do not assume that samples from the test domain are available during training, and we want to design a clustering head that can generalize to new unseen domains at inference time.

To obtain this goal, we present a clustering head training scheme for multi-domain assignments. First, we bridge domain gaps using a Basic Common Domain (BCD), transforming images into a sketch-like domain that retains object identity while reducing biases from color or style. The clustering head is then trained on strong augmentations and style-transferred images. Next, we generate reliable pseudo-labels based on the clustering head's predictions in the BCD (logits) $l(y|x^{bcd}) = f_c \circ f_e(x^{bcd})$, and the semantic features (embeddings) $u = f_e(x)$ of the original image. To enhance clustering stability, we introduce multiple heads and select the most reliable ones by evaluating their diversification. Moreover, we have presented a prediction-based label smoothing technique that adjusts label uncertainties based on past prediction statistics, promoting robust learning across domains. Our holistic strategy ensures effective clustering in high-dimensional image data while addressing the challenges of domain gaps and instability inherent in the clustering process. The steps involved in this method are detailed below, and the training procedure is illustrated in Figure 3.

1. *Data augmentation for clustering head training:* We leverage a BCD to mitigate the gap between several domains. The BCD is designed to maintain the content of each sample while removing domain-related information. A sketch-like domain can be considered a suitable BCD for image data with varying color and texture domains. Transforming an image to a sketch domain keeps high-level features such as object identity while decreasing the bias induced by color or style-preserving features. The clustering head is trained by sampling a batch of $B$ images from all source domains. Each sample $x_i$ passes through $f_e$ in three different versions. Based on the original image, and using two transformations: a strong augmentation $x_i^s = \mathcal{E}(x_i)$, and style transfer to our BCD $x_i^{bcd} = \mathcal{C}(x_i)$. Where $\mathcal{C}()$ represents a style transfer (see the Pre-training section) of the input image $x_i$ to an image with a sketch-like style (Huang & Belongie, 2017). $\mathcal{E}()$ is defined as:

$$x_i^{st} = \begin{cases} \mathcal{C}(x_i) & , \quad w.p \quad p_{st}p_{bcd} \\ \mathcal{ST}(x_i), & w.p \quad p_{st}(1 - p_{bcd}), \\ \mathcal{S}(x_i) & , \quad w.p \quad 1 - p_{st}. \end{cases}$$

The BCD transformed and the original images are used to define the pseudo labels, while the strong augmentations are used to train the clustering head $f_c$, as detailed below.

2. *Reliable pseudo label generation:* First, features are extracted from the original image: $u = f_e(x, \theta_1)$. Then, logits are extracted in the BCD based on: $l(y|x^{bcd}) = f(x^{bcd}) = f_c \circ f_e(x^{bcd})$. The top $\gamma := \frac{B}{2K}$ samples are chosen from $l(y|x_i^{bcd})$ as the set of most confident samples of class $k$ based on the clustering head's score on samples from the BCD. Thus, the selected samples are denoted as follows:

$$\mathcal{M}_k = \{u_i | i \in argtop\gamma(l(k|x^{bcd})), \forall i \in \{1, ..., B\}\}, \tag{2}$$

where $argtop\gamma(l(k|x^{bcd})) \in \mathbb{N}^\gamma$ is a vector of indexes that chooses the $\gamma$ most confident samples, based on their corresponding BCD score $l(k|x^{bcd})$. $\mathcal{M}_k$ is a set of $\gamma$ embedding vectors. Using $M_k$, the center of class $k$ is determined by:

$$G_k = \frac{1}{\gamma} \sum_{u_i \in \mathcal{M}_k} u_i. \tag{3}$$

One can calculate the similarity between each sample and each of the centers: $sim_k = \langle \bar{G}_k, \bar{u} \rangle$. Where $\bar{G}_k = \frac{G_k}{\|G_k\|}$, and $\bar{u} = \frac{u}{\|u\|}$ are the normalized feature and center vectors, respectively. Specifically, $sim_k \in \mathbb{R}^B$ represents how close each sample in the batch is to the center of cluster $k$.

Samples that are closest to the center are used as pseudo labels for cluster $k$; thus, the set of strongly augmented data samples with pseudo labels is formulated as follows:

$$\hat{\mathcal{Z}}_k = \{x^s[argtop\gamma(sim_k)], k\}. \tag{4}$$

Where $x^s[argtop\gamma(sim_k)]$ means indexing $x^s$ using $argtop\gamma(sim_k)$, i.e., choosing the samples that will be used as pseudo-labels for class $k$ using the similarity in the embedding domain. While $M_k$ are chosen based on the predictions in the logit space, $\hat{\mathcal{Z}}_k$ are pseudo labels based on information from both the semantic space Eqs. 3,4 and in the logits space Eq. 2. Using only the logit values to infer the pseudo labels results in poor cluster assignments, as demonstrated in our ablation study. The entire set of representatives and pseudo-label pairs will be denoted as $\hat{\mathcal{Z}}$.

3. *Clustering head training loss:* In the second phase, the clustering head $f_c$ is trained using $\hat{\mathcal{Z}}$ to minimize cross-entropy loss using the pseudo labels. A batch of samples and pseudo labels $(x_{pl}^s, y_{pl}) \in \hat{\mathcal{Z}}$, are propagated through $f_c$, and $f_e$, and the objective can be formulated as:

$$\mathcal{L} = \frac{1}{B} \mathcal{L}_{ce}(f_c \circ f_e(x_{pl}), y_{pl}). \tag{5}$$

During this training phase, domain balancing is used, as detailed in our pre-training section.

4. *Prediction-based label smoothing:* Label smoothing (Szegedy et al., 2016) prevents a classifier from being overconfident, which may lead the classifier to collapse to specific classes and neglect the rest. Using past prediction statistics, we present an improved label smoothing that varies the smoothing intensity between clusters. We perform stronger smoothing for frequently predicted clusters and vice versa. Specifically, let $\hat{y}(x) \in \{0, 1\}^K$ be the one-hot prediction of sample $x$, and we denote the empirical density of predictions over the batch as $\mathcal{H}_{cur} \in \mathbb{R}^K$. We further compute the exponential moving average of prediction density as $\mathcal{H} = \alpha * \mathcal{H} + (1 - \alpha)\mathcal{H}_{cur}$ with $\alpha \in (0, 1)$ is a constant for balancing between history and current statistics. We perform our cluster-specific prediction-based label smoothing using $\mathcal{H}$. Precisely, for cluster $k \in K$ we define the smoothed pseudo-label as $\bar{y}_{pl}[k] = \max(1 - \mathcal{H}[k], s_{min})$, where $s_{min}$ is the minimal smoothing value, and the values of the remaining clusters are spread uniformly to sum up to 1.

5. *Multiple clustering heads:* Clustering is inherently unstable, especially when dealing with many classes or high-dimensional datasets. Several authors have proposed using feature selection (Solorio-Fernández et al., 2020; Shaham et al., 2022; Lindenbaum et al., 2021) to improve clustering capabilities by removing nuisance features in tabular data. We are interested in stabilizing clustering performance on diverse high-dimensional image data. Therefore, we propose training multiple clustering heads simultaneously and selecting a reliable head based on an unsupervised criterion. This allows us to handle many categories and overcome the instability that stems from random weight initialization. For more details about the source of randomization between heads, please see Appendix B.3. The number of clustering heads is denoted as $h$, hence the objective in Eq. 5, $\mathcal{L}$ can now be formulated as the average of the $h$ head specific losses:

$$\mathcal{L} = \frac{1}{Bh} \sum_{i=1}^{h} \mathcal{L}_{ce}^i(f_c \circ f_e(x_{pl}), y_{pl_s}). \tag{6}$$

Next, we define the *diversification* of head $j$ as:

$$dv_j = |\{argmax_{k \in K} l_j(y|x)/K\}|. \tag{7}$$

First, $argmax$ reduces the prediction $l_j(y|x)$ of the $j$-th head to a cluster index. Next, we use the operator $|.|$, which counts the number of distinct cluster predictions in the set of predictions by head $j$; this process is performed over the entire dataset without any parameter update.

Due to high variability in the training procedure between heads, some are better than others; we leverage this variability by keeping only the most diversified heads (MDH). Two MDH are chosen out of $h$ clustering heads based on higher $dv_j$ values compared to the other heads. The heads with lower $dv_j$ are discarded, and we replace the weights of the non-MDH with a linear combination of the two MDH weights. Mathematically speaking, let us define $\theta_{2_i}$ as the weights of the $i$-th head,

and let us assume that $j, k$ are the MDH indices; hence the weights of the non-MDH heads are overridden in the following manner:

$$\theta_{2_i} = r_k \theta_{2_k} + r_j \theta_{2_j}, \forall i \neq k, j. \tag{8}$$

Where $r_k \sim \mathcal{U}(0, 1)$ and $r_j = 1 - r_k$. Inspired by recent weight averaging works (Jain et al., 2023; Yin et al., 2023; Wortsman et al., 2022), this removes the influence of non-diverse heads and maintains some degree of variability for the following optimization steps.

In cases where there is equality in $dv_j$ between several heads, which results in more than two MDHs, we limit the number of MDHs to five. The rationale behind this limitation can be elucidated through the following illustrative scenario, w.l.o.g., assume that the first head does not predict one class, and the other heads do not predict five classes; if the number of MDH kept is not limited, the advantage of the first head is not utilized. Since all heads will perform poorly in the early training phase, MDH selection is initiated after a few epochs. Furthermore, to allow the heads to make gradual learning, the process repeats every $n$ epochs.

## 4 Experiments

Experiments are conducted using four datasets commonly used to evaluate domain generalization methods. Representative images from several datasets and domains appear in Appendix C. **Office31** dataset (Saenko et al., 2010) consists of images collected from three domains: Amazon, Webcam, and DSLR, with 2817, 795, and 498 images, respectively. The dataset includes 31 different classes shared across all domains. The samples consist of objects commonly encountered in an office setting. **PACS** dataset (Li et al., 2017) consists of four domains: Sketch, Cartoon, Photo, and Artpainting with 3929, 2344, 1670, and 2048 images, respectively. It includes seven different classes, which are shared across all domains. **Officehome** dataset (Venkateswara et al., 2017) contains four domains: Art, Product, Realworld, and Clipart, with 2427, 4439, 4357, and 4365 images, respectively. It includes 65 different classes, which are shared across all domains. The large number of domains and classes makes the task challenging. In particular, since we aim to cluster the data without access to labeled observations. Existing state-of-the-art results on this data (Menapace et al., 2020) corroborate this claim. **DomainNet** dataset (Peng et al., 2019), consists of six domains: Real, Painting, Sketch, Clipart, Infograph, and Quickdraw. It includes about 0.6 million images (48K - 172K per domain) distributed among 345 classes.

**Baselines** To evaluate the capabilities of our model, we focus on the following scheme: train the model using $d$ unlabelled source domains, then evaluate our model on the unseen and unlabelled target domain. We compare our approach to several recently proposed deep learning-based clustering models for multi-domain data. Our implementation details appear in Appendix B.1.

When evaluating Office31 and Officehome datasets, we compare with ACIDS (Menapace et al., 2020), Invariant Information Clustering for Unsupervised Image Classification and Segmentation (IIC) (Ji et al., 2018), and DeepCluster (Caron et al., 2018). Importantly, they were trained directly on the target domain before predicting the clusters. We further compare two variations of IIC, specifically, IIC-Merge involves training IIC on all domains, including the target domain; IIC+DIAL: IIC, which contains a domain-specific batch norm layer jointly trained on all domains. We further compare to continuous domain adaptation (Continuous DA) (Mancini et al., 2019), which trains on $d$ domains, then adapted and tested on the target domain. Note that all the former

| Method | Target fine-tuned | Supervision | $D, W \rightarrow A$ | $A, W \rightarrow D$ | $A, D \rightarrow W$ | Avg |
|---|---|---|---|---|---|---|
| DeepCluster (Caron et al., 2018) | ✓ | - | 19.6 | 18.7 | 18.9 | 19.1 |
| IIC (Ji et al., 2018) | ✓ | - | 31.9 | 34.0 | 37.0 | 34.3 |
| IIC-Merge (Ji et al., 2018) | ✓ | - | 29.1 | 36.1 | 33.5 | 32.9 |
| IIC + DIAL (Ji et al., 2018) | ✓ | - | 28.1 | 35.3 | 30.9 | 31.4 |
| Continuous DA (Mancini et al., 2019) | ✓ | - | 20.5 | 28.8 | 30.6 | 26.6 |
| ACIDS (Menapace et al., 2020) | ✓ | - | **33.4** | 36.1 | 37.5 | 35.6 |
| $K$-means (Lloyd, 1982) | ✓ | - | 14.9 | 24.3 | 20.8 | 29.9 |
| Ours | - | - | 24.1 | **50.1** | **47.7** | **40.6** |

Table 1: Accuracy results on the Office31 dataset (31 classes) upon all three domain combinations, each of the letters $A, W, D$ represent the domains Amazon, Webcam, and DSLR, respectively. The notation $X, Y \rightarrow Z$, means the model was trained on $X, Y$ domain and tested on the $Z$ domain. Target fine-tuned means the method was trained or adapted to the test domain. In $K$-means, we first pre-trained the MocoV2 model and trained $K$-means on top of its embeddings.

| Method | Target fine-tuned | Supervision | $C, P, R \rightarrow A$ | $A, P, R \rightarrow C$ | $A, C, R \rightarrow P$ | $A, C, P \rightarrow R$ | Avg |
|---|---|---|---|---|---|---|---|
| DeepCluster (Caron et al., 2018) | ✓ | - | 8.9 | 11.1 | 16.9 | 13.3 | 12.6 |
| IIC (Ji et al., 2018) | ✓ | - | 12.0 | 15.2 | 22.5 | 15.9 | 16.4 |
| IIC-Merge(Ji et al., 2018) | ✓ | - | 11.3 | 13.1 | 16.2 | 12.4 | 13.3 |
| IIC + DIAL(Ji et al., 2018) | ✓ | - | 10.9 | 12.9 | 15.4 | 12.8 | 13.0 |
| Continuous DA (Mancini et al., 2019) | ✓ | - | 10.2 | 11.5 | 13.0 | 11.7 | 11.6 |
| ACIDS (Menapace et al., 2020) | ✓ | - | 12.0 | 16.2 | 23.9 | 15.7 | 17.0 |
| $K$-means (Lloyd, 1982) | ✓ | - | 9.1 | 11.3 | 13.8 | 10.6 | 11.2 |
| Ours | - | - | **20.8** | **26.2** | **27.7** | **27.2** | **25.5** |

Table 2: Results on Officehome dataset (65 classes) upon all four domain combinations, each of the letters $A, P, R, C$ represent the domains Art, Product, Realworld, and Clipart, respectively. The notation $W, X, Y \rightarrow Z$, means the model was trained on $W, X, Y$ domain and tested on the $Z$ domain. Target fine-tuned means the method was trained or adapted to the test domain. In $K$-means, we pre-train the MocoV2 model and then train $K$-means on top of its embeddings.

baselines compared with our work were trained on the target domain. We added another baseline, training MoCoV2 on all the source domains and fitting the $K$-means clustering algorithm on the target domain.

On the PACS dataset, we also compared to BrAD (Harary et al., 2022) with various amounts of source domain labels. This comparison is very challenging as we do not use any class labels. Evaluation on DomainNet (Peng et al., 2019) was done following the protocol for UDG introduced in Zhang et al. (2021). Our method does not utilize labels compared to other methods, which use 1% source domain labels.

**Results** Table 1 depicts the results on all three domain combinations of the Office31 dataset. Our method outperforms the current state-of-the-art (SOTA) by a large margin on both DSLR and Webcam as target domains, even without adaptation to the target domain. However, our method's performance on the Amazon domain is inferior to the current SOTA, which may be due to limited source domain data. The target fine-tuned method, which relies on unsupervised fine-tuning on the target domain, uses 317% more data for their training scheme. On average, our method performs better by 10.1% than methods that use the target domain for adaptation and 31.1% over baselines with the same conditions.

Results on the Officehome dataset can be seen in Table 2. This dataset is more challenging than the former and consists of four domains. Our method outperforms the baselines on all four domain combinations and is better on average by 47.1% than the previous SOTA.

On the PACS dataset (Table 3), we compare to target fine-tuned and limited source domain label methods. Our method outperforms the current SOTA on 3 out of 4 target domains for the target fine-tuned case. Our method achieves slightly lower results on the fourth domain (Sketch) than the current target fine-tuned SOTA. This can be explained by a large amount of additional Sketch domain data (65%), exploited by baselines fine-tuned on the target domain. On average, our method outperforms all baselines that do not require any level of supervision. When comparing the two variations of BrAD with 1% source domain labels, we achieve superior results on all domains. Overall, our method outperforms BrAD-KNN, even though BrAD-KNN uses 10% of the source domain labels.

Table 4 illustrates the results on DomainNet dataset (Peng et al., 2019). Our method is compared to different SOTA UDG methods that use 1% of source domain labels. Although our method does not use any class labels, it outperforms all previous schemes across all domains by a large margin.

**Ablation study** We conducted two ablation studies to evaluate our model's performance. The first study explores different variations of our model, while the second study focuses on the proposed

| Method | Target fine-tuned | Supervision | $C, P, S \to A$ | $A, P, S \to C$ | $A, C, S \to P$ | $A, C, P \to S$ | Avg |
|---|---|---|---|---|---|---|---|
| DeepCluster(Caron et al., 2018) | ✓ | - | 22.2 | 24.4 | 27.9 | 27.1 | 25.4 |
| IIC (Ji et al., 2018) | ✓ | - | 39.8 | 39.6 | 70.6 | 46.6 | 49.1 |
| IIC-Merge (Ji et al., 2018) | ✓ | - | 32.2 | 33.2 | 56.4 | 30.4 | 38.1 |
| IIC + DIAL(Ji et al., 2018) | ✓ | - | 30.2 | 30.5 | 50.7 | 30.7 | 35.3 |
| Continuous DA (Mancini et al., 2019) | ✓ | - | 35.2 | 34.0 | 44.2 | 42.9 | 39.1 |
| ACIDS (Menapace et al., 2020) | ✓ | - | 42.1 | 44.5 | 64.4 | **51.1** | 50.5 |
| $K$-means (Lloyd, 1982) | ✓ | - | 17.7 | 18.5 | 21.1 | 22.4 | 19.9 |
| Ours | - | - | **47.3** | **45.4** | **66.6** | 48.0 | **51.8** |
| BrAD (Harary et al., 2022) | - | 1% | 33.6 | 43.5 | 61.8 | 36.4 | 43.8 |
| BrAD-KNN (Harary et al., 2022) | - | 1% | 35.5 | 38.1 | 55.0 | 34.1 | 40.7 |
| BrAD (Harary et al., 2022) | - | 5% | 41.4 | 50.9 | 65.2 | 50.7 | 52.0 |
| BrAD-KNN (Harary et al., 2022) | - | 5% | 39.1 | 45.4 | 58.7 | 46.1 | 47.3 |
| BrAD (Harary et al., 2022) | - | 10% | 44.2 | 50.0 | 72.2 | 55.7 | 55.5 |
| BrAD-KNN (Harary et al., 2022) | - | 10% | 42.0 | 45.3 | 67.2 | 50.0 | 51.1 |

Table 3: Results on PACS dataset (7 classes) upon all four domain combinations, each letter $A, P, S, C$ represents the domains: Art painting, Photo, Sketch, and Cartoon, respectively. The notation $W, X, Y \to Z$, means the model was trained on $W, X, Y$ domain and tested on the $Z$ domain. Target fine-tuned means the method was trained or adapted to the test domain. In $K$-means, we first pre-trained the MocoV2 model and trained $K$-means on top of its embeddings.

| Method | Supervision | clipart, infograph, quickdraw | | | painting, real, sketch | | | Avg |
|---|---|---|---|---|---|---|---|---|
| | | $\to$ painting | $\to$ real | $\to$ sketch | $\to$ clipart | $\to$ infograph | $\to$ quickdraw | |
| DIUL (Zhang et al., 2021) | 1% | 14.5 | 21.7 | 21.3 | 18.5 | 10.6 | 12.7 | 16.6 |
| DiMEA (Yang et al., 2022) | 1% | 20.2 | 30.8 | 20.0 | 26.5 | 15.5 | 15.5 | 21.4 |
| BrAD-KNN (Harary et al., 2022) | 1% | 16.9 | 22.3 | 25.7 | 40.7 | 14.0 | 21.3 | 23.5 |
| BrAD (Harary et al., 2022) | 1% | 20.0 | 25.1 | 31.7 | 47.3 | 16.9 | 23.7 | 27.5 |
| Ours | - | **27.0** | **29.0** | **40.0** | **52.2** | **17.4** | **32.4** | **33.0** |

Table 4: Accuracy results on DomainNet dataset (Peng et al., 2019) upon two source domain combinations. The notation $X, Y \to Z$, means the model was trained on $X, Y$ domain and tested on the $Z$ domain. Without using any labels, our results outperform other SOTA UDG methods, which use 1% of source domain labels.

| Method | $C, P, S$ $\to A$ | $A, P, S$ $\to C$ | $A, C, S$ $\to P$ | $A, C, P$ $\to S$ | Avg |
|---|---|---|---|---|---|
| Logits only | NA | NA | NA | NA | NA |
| Plain pre-training | 25.0 | 22.7 | 29.8 | NA | NA |
| No style transfer | 40.6 | 37.2 | 50.2 | 45.8 | 43.5 |
| No domain head | 40.2 | 41.5 | 58.6 | 42.8 | 45.8 |
| No smoothing | 48.2 | 44.5 | 56.8 | 46.6 | 49.0 |
| Label smoothing | 46.3 | 44.4 | **66.6** | **49.0** | 51.6 |
| PB label smoothing | **47.3** | **45.4** | **66.6** | 48.0 | **51.8** |

Table 5: Ablation study on the PACS dataset. The letters $A, P, S$, and $C$ represent the domains Art painting, Photo, Sketch, and Cartoon, respectively. The arrow notation is similar to other tables. We denote NA cases in which some of the classes were not predicted by the model, which makes calculating clustering accuracy unavailable.

prediction-based (BP) label smoothing. In the first study, we used the PACS dataset and tested four variations of our model. The first variant, called "Plain pre-training", used a standard MoCoV2 feature extractor, followed by training the clustering heads. In the second variation, we omitted the style transfer augmentation during pre-training and clustering head training. The third variation, "no domain head," excluded the domain head and its adversarial loss from the entire training procedure. The fourth variation removed the label smoothing, using a smoothing value of 1. The results presented in Table 5 showed that our model performed better than all its ablated versions. This suggests that all the proposed components contribute to our ability to generalize to unseen domains. In cases where not all clusters were predicted, and thus, no clustering accuracy could be calculated, the result was denoted as NA.

In the second ablation study, we examined three variations of label smoothing: no smoothing, standard label smoothing, and prediction-based label smoothing. We evaluated our model's performance on the PACS, Officehome, and Office31 datasets. Our results showed that the proposed prediction-based label smoothing improved clustering capabilities across all evaluated datasets. We presented the results on PACS in Table 5, while the results on the other datasets are included in Appendix B.4. We excluded the results indicating the "single head" ablated version of our model because its prediction often did not cover all clusters.

**Hyperparameter stability** We use PACS and Officehome datasets to evaluate our model's sensitivity to hyperparameters. We test three hyperparameters and present our results in Appendix B.5 suggesting that our method is nearly insensitive to slight variations in the hyperparameters.

## 5 Discussion

**Conclusion:** We have developed a novel framework for clustering that is fully unsupervised and can be applied to multiple domains. Our approach does not rely on class labels from either the source or target domains, which is a significant advantage. Moreover, our method does not require adaptation to the target domain, making it more capable of generalizing to new, previously unseen domains. We have compared our approach to existing baselines and found that it outperforms them. Furthermore, we plan to extend our model to other modalities, such as audio and text, and apply it to other unsupervised learning tasks, such as feature selection or anomaly detection. We are currently working on a theoretical analysis of our results. We believe that our framework

has great potential to advance the unsupervised multi-domain regime and can be used for future research.

**Limitations**   We should note that our method has certain limitations when handling datasets with a large number of classes. However, one possible solution to mitigate this issue is to use weak supervision. Moreover, we acknowledge that our current implementation cannot accommodate non-overlapping classes between the training and testing subsets. This presents an interesting question for future research that needs to be addressed.

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

# A    Appendix

# B    Additional implementation details and results

## B.1    Implementation details

Our work is implemented on an NVIDIA RTX 3080 GPU using PyTorch (Paszke et al., 2019). We use blank Resnet18 (He et al., 2016) as our feature extractor to align with the baseline (Menapace et al., 2020; Harary et al., 2022; Wang et al., 2022) The models in the first phase were trained using SGD with momentum 0.9 and weight decay $1e - 4$. We use a batch size of 8 and train the model for 500 epochs. To train the clustering head, we use the same optimizer with batches of size 256 for 100 epochs for Office31 and Officehome datasets and 50 epochs for the PACS dataset. This difference is due to the small number of classes in the PACS dataset, which enables the model to converge much faster. To create style transfer augmentations, we use a pre-trained AdaIN model (Huang & Belongie, 2017). The most diversified head selection mechanism initiates at epoch 30 and is repeated every $n = 10$ epochs. For more information on the head selection mechanism, see the Multiple Clustering Heads section in the main text of our work. An important regularization for diversified training is label smoothing (Szegedy et al., 2016). Using pseudo-labels, we assume a high ratio of mislabeled samples; label smoothing helps prevent the model from predicting the training samples too confidently. The ablation study shows empirical evidence of the importance of label smoothing in our task.

## B.2    Implementation of strong augmentations

In the pre-training phase 3.1, we perform style transfer or strong augmentations for each batch. When a batch is chosen to undergo strong augmentations, we use multiple augmentations which a chained together, creating the strong augmentation. Each augmentation component is chosen by a random probability. Those components include: random resized crop with a usage probability of $p = 0.8$, color jitter ($p = 0.8$), and parameters brightness=0.4, contrast=0.4, saturation=0.4, hue=0.1. Additional components are random grayscale ($p = 0.2$), horizontal flip ($p = 0.5$), and Gaussian blur ($p = 0.5$). Strong augmentation for the clustering head training 3.2 was done with the same strategies used in SCAN (Van Gansbeke et al., 2020): Cutout (DeVries & Taylor, 2017) and four randomly transformations from RandAugment (Cubuk et al., 2020).

## B.3    Source of randomization between heads

Each head weight is initialized randomly using PyTorch (Paszke et al., 2019) default initialization. Since the classifier prediction determines the pseudo labels, each head will be trained using different pseudo labels; this variability will keep propagating as training proceeds.

## B.4    Additional ablation studies

This section presents the results of an ablation study performed on three datasets. Tables 6,7,8 show results of our method without any smoothing, with label smoothing, and using prediction-based label smoothing. In most cases, our proposed prediction-based label smoothing is the best-performing method, illustrating its contribution to the overall method.

Table 6: Different label smoothing techniques ablation study results on the Officehome dataset (Venkateswara et al., 2017).

| Method | $C, P, R \to A$ | $A, P, R \to C$ | $A, C, R \to P$ | $A, C, P \to R$ | Avg |
|---|---|---|---|---|---|
| No smoothing | 20.6 | 25.2 | 27.1 | 27.3 | 25.0 |
| Label smoothing | 20.8 | 25.5 | 27.9 | 25.6 | 25.0 |
| PB label smoothing | **20.8** | **26.2** | **27.7** | **27.2** | **25.5** |

Table 7: Different label smoothing techniques ablation study results on the PACS dataset (Li et al., 2017).

| Method | $C, P, S \to A$ | $A, P, S \to C$ | $A, C, S \to P$ | $A, C, P \to S$ | Avg |
|---|---|---|---|---|---|
| No smoothing | 48.2 | 44.5 | 56.8 | 46.6 | 49.0 |
| Label smoothing | 46.3 | 44.4 | **66.6** | **49.0** | 51.6 |
| PB label smoothing | **47.3** | **45.4** | **66.6** | 48.0 | **51.8** |

Table 8: Different label smoothing techniques ablation study results on the Office31 dataset (Saenko et al., 2010).

| Method | $D, W \to A$ | $A, D \to W$ | $A, W \to D$ | Avg |
|---|---|---|---|---|
| No smoothing | 24.0 | 50.0 | 47.4 | 40.4 |
| Label smoothing | 23.1 | 49.2 | 45.2 | 39.2 |
| PB label smoothing | **24.1** | **50.1** | **47.7** | **40.6** |

### B.5 Sensitivity to hyperparameters

We perform a hyperparameter stability sensitivity study on different hyperparameters. We show that significant changes in hyperparameters do not lead to major changes in the performance of our proposed method. Specifically, we show a deviation of less than 6% in performance for all tested hyperparameter changes (see Table 9).

Table 9: Hyperparameter stability results-$p_{st}$ is the probability of doing style augmentation on the input data. $p_{bcd}$ is the probability of transforming the input sample to the BCD, and $\#MDH$ is the number of cluster heads to keep during training.

| Dataset | Variation | Hyper-parameter | Base value | Deviated value | Base Acc | Deviated Acc | Deviation |
|---|---|---|---|---|---|---|---|
| PACS | $A, P, S \rightarrow C$ | $p_{bcd}$ | 0.2 | 0.3 | 44.7 | 44.1 | 1.34% |
| PACS | $A, P, S \rightarrow C$ | $p_{bcd}$ | 0.2 | 0.4 | 44.7 | 46.1 | 3.13% |
| PACS | $A, P, S \rightarrow C$ | $p_{st}$ | 0.3 | 0.2 | 44.7 | 45.0 | 0.67% |
| PACS | $A, P, S \rightarrow C$ | $p_{st}$ | 0.3 | 0.4 | 44.7 | 45.6 | 2.01% |
| PACS | $A, P, S \rightarrow C$ | MDHs | 5 | 3 | 44.7 | 46.2 | 3.35% |
| PACS | $A, P, S \rightarrow C$ | MDHs | 5 | 7 | 44.7 | 46.6 | 4.25% |
| PACS | $A, C, S \rightarrow P$ | $p_{bcd}$ | 0.2 | 0.3 | 66.6 | 64.8 | 2.70% |
| PACS | $A, C, S \rightarrow P$ | $p_{bcd}$ | 0.2 | 0.4 | 66.6 | 66.2 | 0.60% |
| PACS | $A, C, S \rightarrow P$ | $p_{st}$ | 0.3 | 0.2 | 66.6 | 62.8 | 5.71% |
| PACS | $A, C, S \rightarrow P$ | $p_{st}$ | 0.3 | 0.4 | 66.6 | 63.6 | 4.50% |
| PACS | $A, C, S \rightarrow P$ | MDHs | 5 | 3 | 66.6 | 63.1 | 5.26% |
| PACS | $A, C, S \rightarrow P$ | MDHs | 5 | 7 | 66.6 | 65.3 | 1.95% |
| Officehome | $A, C, R \rightarrow P$ | $p_{bcd}$ | 0.2 | 0.3 | 27.7 | 28.5 | 2.89% |
| Officehome | $A, C, R \rightarrow P$ | $p_{bcd}$ | 0.2 | 0.4 | 27.7 | 26.8 | 3.25% |
| Officehome | $A, C, R \rightarrow P$ | $p_{st}$ | 0.3 | 0.2 | 27.7 | 27.9 | 0.72% |
| Officehome | $A, C, R \rightarrow P$ | $p_{st}$ | 0.3 | 0.4 | 27.7 | 27.8 | 0.36% |
| Officehome | $A, C, R \rightarrow P$ | $p_{st}$ | 0.3 | 0.4 | 27.7 | 27.8 | 0.36% |
| Officehome | $A, C, R \rightarrow P$ | MDHs | 5 | 3 | 27.7 | 28.3 | 2.16% |
| Officehome | $A, C, R \rightarrow P$ | MDHs | 5 | 7 | 27.7 | 27.6 | 0.36% |
| Officehome | $A, C, P \rightarrow R$ | $p_{bcd}$ | 0.2 | 0.3 | 27.2 | 27.4 | 0.74% |
| Officehome | $A, C, P \rightarrow R$ | $p_{bcd}$ | 0.2 | 0.4 | 27.2 | 27.4 | 0.74% |
| Officehome | $A, C, P \rightarrow R$ | $p_{st}$ | 0.3 | 0.2 | 27.2 | 27.0 | 0.74% |
| Officehome | $A, C, P \rightarrow R$ | $p_{st}$ | 0.3 | 0.4 | 27.2 | 25.6 | 5.88% |
| Officehome | $A, C, P \rightarrow R$ | MDHs | 5 | 3 | 27.2 | 27.6 | 1.47% |
| Officehome | $A, C, P \rightarrow R$ | MDHs | 5 | 7 | 27.2 | 27.7 | 1.84% |

## C  Sample images

Figure 4 presents example images from the original domains and our BCD. Next, In Figure 5 we present sample images from datasets used in our paper.

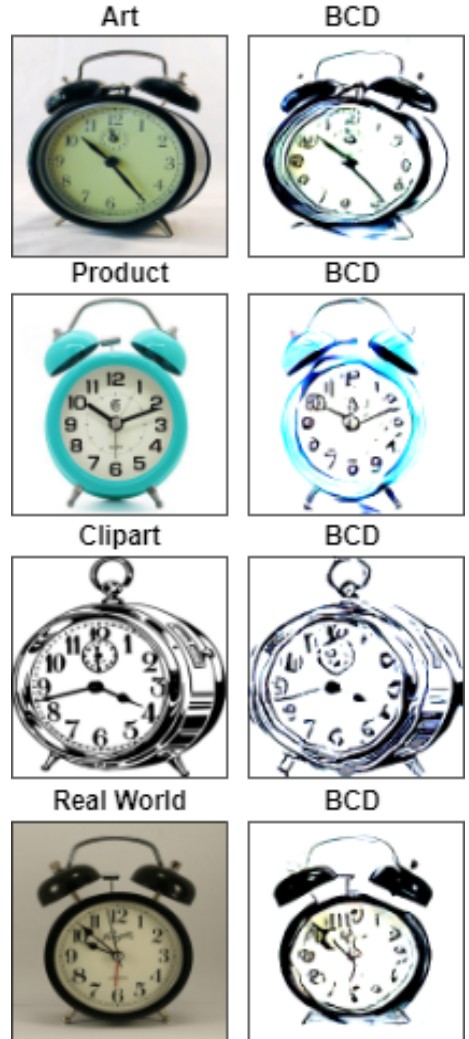

Figure 4: Sample images from BCD domain for Officehome dataset (Venkateswara et al., 2017). The right and left columns show the original image and its BCD transform.

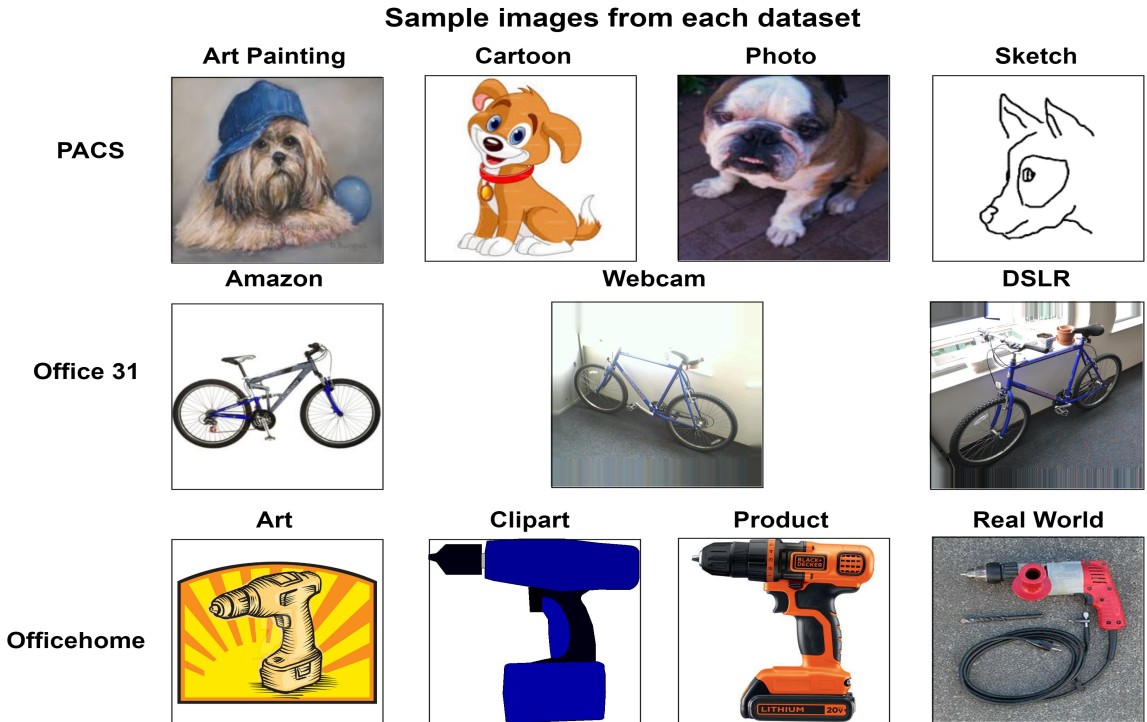

Figure 5: Sample images from the datasets used in the paper.

