# OpenReview forum: "Domain-Generalizable Multiple-Domain Clustering"
_TMLR — Accepted by TMLR_

### Review · Reviewer_y9f5 · 2023-12-13

**Summary Of Contributions:**

Summary:
The paper presents a novel task setting in which no labeled samples are available in the source domain, and the target domain is unseen during the training stage.
To address this task, the paper introduces a two-stage training framework. The first stage involves self-supervised pre-training, while the second stage focuses on multi-head cluster prediction. Experimental evaluations are conducted on standard benchmarks.

**Audience:**

Yes

**Broader Impact Concerns:**

Nan.

**Claims And Evidence:**

Yes

**Requested Changes:**

See weaknesses

**Strengths And Weaknesses:**

Strengths:
1. A potentially new task setting. The explored task setting differs from previous studies and holds practical significance.
2. The proposed method consists of two training stages, which involve learning domain-invariant representations and cluster prediction, respectively.


Weaknesses:

1. The technical novelty appears to be somewhat limited. Several techniques, such as style transfer augmentation, domain adversarial training, and domain-specific queues, are referenced from other papers. As a result, the proposed method incorporates many existing techniques from other methods, and the true technical contribution is unclear.
2. The definition of UDA, DG, and UDG is unclear. To the best of my knowledge, UDA and DG are different, with UDA additionally utilizing unlabeled target data. However, this distinction is not depicted in Figure 1.
3. Pre-trained baseline models like CLIP have demonstrated strong generalization capabilities across domains. Therefore, I am curious about the performance comparison between the proposed method and these pre-trained models. Such a comparison could further validate the effectiveness of the proposed method.

---

> ### Author Response · Authors · 2023-12-21
>
> We thank reviewer y9f5 for appreciating the quality and novelty of our work. In the following, we address all points raised by the reviewer.
>
> 1. We appreciate the feedback regarding the perceived lack of significant novel elements in our framework. We would like to address this comment by highlighting the contributions of our work, which goes beyond the technical algorithmic components:
>
>         1.1 Formulate a new learning task in which no labeled samples are available in the source domain, and the target domain is unseen during training.
>         1.2 Introduce a two-stage training scheme that first learns semantically meaningful features and then predicts cluster assignments while generalizing to new unseen domains.
>         1.3 Present a two-stage training scheme for the proposed interpretable clustering problem. Besides the integrations of existing methodologies, we have also introduced new components. These include (i) a special pseudo-label selection procedure that exploits a style-augmented common domain to identify reliable labels. (ii) a new type of unsupervised label smoothing, which we call prediction-based label smoothing, that prevents mode collapse to highly populated clusters and bad initialization. (iii) a multiple clustering head prediction and filtration procedure to stabilize the clustering step in the presence of high-dimensional data or many categories. Our cluster assignment scheme uses both the semantic space and the logits space with multiple domains, which results in better handling multiple head clustering with a large number of clusters.
>         1.4 Extensive empirical evaluation of the model's ability to predict accurate cluster assignments across domains. Including an ablation study demonstrating the importance of each proposed component.
>
> As a final remark about novelty, we want to emphasize that many well-celebrated papers in the ML community rely on the combination and adaptation of existing schemes; examples of such works include [1], [2], [3]. We believe that this should be encouraged by the community. The main novelty here is that the presented problem and solution were never addressed before.
>
> [1] Deep Residual Learning for Image Recognition - used skip connections (which was already an existing technique) to enable gradient propagation in deep layers.
> [2]Attention Is All You Need - used the attention mechanism (commonly applied for text) to image data.
> [3] BERT: Pre-training of Deep Bidirectional Transformers for Language Understanding - used existing transformers but trained to predict text conditioned on both left and right context in all layers.
>
> 2. We have added a row representing DG in the figure and updated the caption to clarify this issue:
> “Different tasks for multi-domain image classification or clustering. Domain adaptation (DA) deals with adapting a model from one domain to another and requires the availability of labeled samples in the source and target domain labels for training the model [4]. Unsupervised domain adaptation (UDA) alleviates the need for labeled target samples and is trained using labeled source samples and unlabeled samples from the target domain [5]. Domain generalization (DG) aims to create a model that generalizes across multiple source and unseen target domains. The training procedure is similar to UDA but does not require access to samples (unlabeled) from the target domain during training [6,7]. Unsupervised domain generalization (UDG) does not assume access to target samples but relies only on a limited amount of labeled data in the source domain [8,9]. Unsupervised clustering under domain shift (UCDS) does not rely on labeled data but requires access to samples from the source and target domains for training [10]. Our fully unsupervised approach can generalize to the target domain without access to its samples during training.”
>
> [4] Csurka et al. [5] Piva et al. [6] Roy et al. [7] Matsuura et al. [8] Harary et al. [9] Xingxuan et al. [10] Menapace et al.
>
> 3. The main goal of our work is to develop a method for training a NN to predict cluster assignment on multiple domain data without access to any labeled examples. CLIP (Contrastive Language-Image Pre-training) is not an unsupervised model; rather, it learns powerful visual and textual representations by training on a huge dataset (>400M examples) where images and textual descriptions are paired, providing a very strong form of supervision. In contrast, our model is fully unsupervised and is trained without any paired examples. For these reasons, we argue that comparing our model to CLIP would not lead to any valuable insights regarding the capabilities of performing the proposed task. Furthermore, in real-world use cases, such as clustering of medical images [11], no paired examples are available. CLIP is not applicable, but our model could be valuable for performing accurate clustering across domains.
>
> [11] A new clustering method for the diagnosis of CoVID19 using medical images.

---

### Review · Reviewer_rk48 · 2023-12-19

**Summary Of Contributions:**

This paper studies the Domain Generalization problem, where only unlabeled data are given without any supervised knowledge. To solve this problem, a two-stage framework is proposed: First, the authors use self-supervised learning conduct representation learning to obtain style consistency. Then, using clustering with pseudo labels, the label prediction could be effectively inferred. By introducing several other training tricks, such as domain balancing, data augmentation, reliable pseudo labels, etc, the authors claim that the performance can be further refined. After evaluating on the effectiveness of the proposed method, the authors show on several typical domain generalization datasets, such as Office-31, PACS, VLCS, and DomainNet, the domain generalization result indeed can be largely improved by the proposed framework.

**Audience:**

Yes

**Broader Impact Concerns:**

No ethical concerns. Limitions are discussed in the appendix.

**Claims And Evidence:**

Yes

**Requested Changes:**

Please see the weaknesses.

**Strengths And Weaknesses:**

Strengths:
1.	This paper is clearly written and easy to follow. I can quickly get the main idea of this paper.
2.	The problem is novel and challenging. Current Domain Generalization is usually conducted under full supervision. But in this paper, the authors study the purely unsupervised setting.
3.	The experimental results are good. For example, in Office-31 W->D and OfficeHome, the performance is improved for over 10%.

Weaknesses:
1.	There is not theoretical guarantee of the proposed method. We cannot be sure that under such framework a machine learning model could be trained to converge to a certain point.
2.	The essential idea of this paper is not novel. There are several similar works using style augmentation that are also close to the field of domain generalization, such as:
- Zhou et al., Domain Generalization with MixStyle.
- Li et al., A Simple Feature Augmentation for Domain Generalization.
- Huang et al., Harnessing Out-Of-Distribution Examples via Augmenting Content and Style.
- Zeng et al., Foresee What You Will Learn: Data Augmentation for Domain Generalization in Non-stationary Environment.
Moreover, there is no discussion of these related works.
3.	There are too many tricks introduced as a methodology contribution which makes the proposed framework very incompact and not easy to implement. Moreover, there is no code submitted for examination of the reproducibility.
4.	Some of the compared methods are outdated. State-of-the-art performed methods such as SWAD: Domain Generalization by Seeking Flat Minima are not included as a baseline method.
5.	There is no computational complexity discussed in the experiments. How much memory is required to store all the negative queues? Is it still applicable when the framework is conducted on large-scale datasets such as LAION-400-milion?

---

> ### Author Response · Authors · 2024-01-02
>
> We thank the reviewer rk48 for appreciating our paper’s writing quality, the novelty of our work, and our empirical results. In the following, we address all points raised by the reviewer.
>
> 1. While our current work is largely experimental, we are also focusing on its theoretical aspects, as discussed in the relevant section. We understand the importance of convergence guarantees, but we also believe that our experimental findings will motivate the research community to explore this crucial topic.
>
> 2. We appreciate the feedback regarding the perceived lack of significant novel elements in our framework. We would like to address this comment by highlighting the contributions of our work, which goes beyond the technical algorithmic components:
>
>     2.1 Formulate a new learning task in which no labeled samples are available in the source domain, and the target domain is unseen during the training stage.
>
>     2.2 Introduce a two-stage training scheme that first learns semantically meaningful features and then predicts cluster assignments while generalizing to new unseen domains.
>
>     2.3 Present a two-stage training scheme for the proposed interpretable clustering problem. Besides the integrations of existing methodologies, we have also introduced new components. These include (i) a special pseudo-label selection procedure that exploits a style-augmented common domain to identify reliable labels. (ii) a new type of unsupervised label smoothing, which we call prediction-based label smoothing, that prevents mode collapse to highly populated clusters and bad initialization. (iii) a multiple clustering head prediction and filtration procedure to stabilize the clustering step in the presence of high-dimensional data or many categories. Our cluster assignment scheme uses both the semantic space and the logits space with multiple domains, which results in better handling multiple head clustering with a large number of clusters.
>
>     2.4 Extensive empirical evaluation of the model's ability to predict accurate cluster assignments across domains. Including an ablation study demonstrating the importance of each proposed component.
>
> As a final remark about novelty, we want to emphasize that many well-celebrated papers in the ML community rely on the combination and adaptation of existing schemes; examples of such works include [1], [2], [3]. We believe that this should be encouraged by the community. The main novelty here is that the presented problem and solution were never addressed before.
>
> [1] Deep Residual Learning for Image Recognition - used skip connections (which was already an existing technique) to enable gradient propagation in deep layers.
>
> [2]Attention Is All You Need - used the attention mechanism (commonly applied for text) to image data.
>
> [3] BERT: Pre-training of Deep Bidirectional Transformers for Language Understanding - used existing transformers but trained to predict text conditioned on both left and right context in all layers.
>
> Thanks for pointing out all these related papers. Indeed, most of these works focus on domain generalization but are supervised and require access to labels from all source domains. We provide here a brief description of each of these works in the updated version of our manuscript in “background and related work” (section 2).
>
> 3. Our submission includes the code in the supplementary material, so it should be easy to reproduce our results. We also plan to release the code to Github. We hope this will help other researchers easily reproduce and build upon it.
> 4. “SWAD: Domain Generalization by Seeking Flat Minima“ is a domain generalization scheme that was tested using training labels and is from 2021. We do compare to more recent SOTA models like [4,5,6].
>
> [4] Harary et al.
>
> [5] Zhang et al.
>
> [6] Yang et al.
>
> 5. Our architecture follows [7] with a ResNet-18 as a backbone for a fair comparison with the other baselines. As mentioned in the implementation details, we used a single NVIDIA RTX 3080 GPU with 12GB, which is relatively modest when compared to recent LLM/LVLM models. The negative queue memory footprint is quite low as it stores embedding vectors (128 dimensions) and not full images. The difference in our first stage between our default queue size of 65536 to a smaller one of 512 is 127MB. The Laion dataset [8] or any of its curated versions can definitely be used for our training. It will consume the same amount of memory as any other data set due to the fixed input image and batch size. In cases with more memory, we recommend using a larger batch size to accelerate the training process.
>
> [7] He et al. Momentum contrast for unsupervised visual representation learning. 2020.
>
> [8] Schuhmann et al. "Laion-400m: Open dataset of clip-filtered 400 million image-text pairs." 2021.

---

### Review · Reviewer_Q7oG · 2023-12-21

**Summary Of Contributions:**

This paper investigates the multi-source domain generalized clustering,  where no class labels are used for either source or target domains. This setting is interesting and difficult. They propose a two-stage training framework: (1) self-supervised pre-training for extracting domain invariant semantic features. (2) multi-head cluster prediction with pseudo labels, further leveraging a novel prediction-based label smoothing scheme. The experimental results are good.

**Audience:**

Yes

**Claims And Evidence:**

Yes

**Requested Changes:**

- Wrtting quality.
- More basline methods.

**Strengths And Weaknesses:**

Strengths
- The proposed method is simple and easy to follow.
- The experimental results are good.

Weaknesses
- The technique of this work lacks novelty. Although the performance seems to be good, their proposed algorithm is like a stack of existing tricks, such as style augmentation, label smoothing, and adversarial domain adaptation.
- According to the method part, the proposed method aims to eliminate the domain differences. Therefore, given a mixed domain of multiple different domains, would this method be still effective?
- The writting has to be further improved. For example, the grammar and the format of reference have some issues. But the writing problem is not limited to what I've pointed out.
- Some important baselines are missing, such as SCAN and DBSCAN.

---

> ### Author Response · Authors · 2024-01-02
>
> We thank reviewer Q7oG for appreciating our work setting and experimental results. In the following, we address all points raised by the reviewer.
>
> 1. We appreciate the feedback regarding the perceived lack of significant novel elements in our framework. We would like to address this comment by highlighting the contributions of our work, which goes beyond the technical algorithmic components:
>
>         1.1 Formulate a new learning task in which no labeled samples are available in the source domain, and the target domain is unseen during the training stage.
>         1.2 Introduce a two-stage training scheme that first learns semantically meaningful features and then predicts cluster assignments while generalizing to new unseen domains.
>         1.3 Present a two-stage training scheme for the proposed interpretable clustering problem. Besides the integrations of existing methodologies, we have also introduced new components. These include (i) a special pseudo-label selection procedure that exploits a style-augmented common domain to identify reliable labels. (ii) a new type of unsupervised label smoothing, which we call prediction-based label smoothing, that prevents mode collapse to highly populated clusters and bad initialization. (iii) a multiple clustering head prediction and filtration procedure to stabilize the clustering step in the presence of high-dimensional data or many categories. Our cluster assignment scheme uses both the semantic space and the logits space with multiple domains, which results in better handling multiple head clustering with a large number of clusters.
>         1.4 Extensive empirical evaluation of the model's ability to predict accurate cluster assignments across domains. Including an ablation study demonstrating the importance of each proposed component.
>
> As a final remark about novelty, we want to emphasize that many well-celebrated papers in the ML community rely on the combination and adaptation of existing schemes; examples of such works include [1], [2], [3]. We believe that this should be encouraged by the community. The main novelty here is that the presented problem and solution were never addressed before.
>
> [1] Deep Residual Learning for Image Recognition - used skip connections (which was already an existing technique) to enable gradient propagation in deep layers.
>
> [2] Attention Is All You Need - used the attention mechanism (commonly applied for text) to image data.
>
> [3] BERT: Pre-training of Deep Bidirectional Transformers for Language Understanding - used existing transformers but trained to predict text conditioned on both left and right context in all layers.
>
> 2. Our method has been evaluated using various datasets, and we have demonstrated its stable generalization capabilities for different domain combinations. Although obtaining the correct domain label based on the data source is usually straightforward, a mixed domain as a source will not significantly impact most aspects of our method. This is because we attenuate the influence of domain-specific information by converting all images to the BCD and using style augmentations with a contrastive loss.
>
> 3. After receiving this feedback, we made significant efforts to proofread and correct our paper. During this process, we improved the phrasing, corrected the citations, and expanded some of our explanations.
>
> 4. Adding the suggested baseline, SCAN, using our settings of training on multiple unlabelled source domains and testing on an unseen target domain:
> |Method|Target fine-tuned|Supervision|C,P,S->A|A,P,S->C|A,C,S->P|A,C,P->S|Avg|
> |-|-|-|-|-|-|-|-|
> |SCAN| V | - |26.6|32|31.8|33.1|30.9|
> |Ours| - | - |47.3|45.4|66.6|48|51.8|
>
> In our approach, we utilized the Resnet18 backbone, which is also used by other methods in our paper, and the same input image size. Our proposed method has an average accuracy of 51.8, whereas SCAN achieves significantly worse results with an average accuracy of 30.9. We believe that the performance gap is mainly due to the design of the SCAN model, which requires the presence of the target domain during all its training stages.
>
> DBSCAN (Density-Based Spatial Clustering of Applications with Noise) is a clustering algorithm that is effective in identifying clusters based on the density of the data. However, DBSCAN's performance can be affected by the curse of dimensionality, which refers to the increased difficulty of organizing and searching high-dimensional data [4,5]. Since our input data is extremely high dimensional, we don’t see how DBSCAN can be evaluated on our benchmarks.
>
> [4] Adil Abdu et ali. "Comparative analysis review of pioneering DBSCAN and successive density-based clustering algorithms." 2021
>
> [5] Changlock et al. "Mdst-dbscan: A density-based clustering method for multidimensional spatiotemporal data." 2021

---

### Decision · Action_Editor_w9MU · 2024-01-27

**Recommendation:** Accept as is

**Comment:**

The reviewers found the problem setting “interesting and difficult” (Reviewer Q7oG), “novel and challenging” (Reviewer rk48), and holds practical significance (Reviewer y9f5). The experimental results are “good” (Reviewer Q7oG, Reviewer rk48), the proposed method is “simple and easy to follow” (Reviewer Q7oG). During the rebuttal, the authors have made modifications and clarifications to the paper, and presented a comparison to an additional baseline that the reviewers mentioned (with their method comparing favourably). I therefore believe the paper meets the bar of acceptance at TMLR.

For the camera ready, I strongly encourage the authors to include qualitative analysis such as feature visualizations that can aid in interpreting the strong performance of the proposed method and strengthen the paper.

**Audience:**

Clustering and unsupervised domain generalization are important and interesting problem settings and the proposed approach would be of interest to members of the community.

**Claims And Evidence:**

This paper presents a framework for clustering across domains. In particular, they are interested in discovering clusters that will work well for unseen domains at test time. Contrary to prior work, they do not use any supervision at any point of their pipeline. Their approach is based on a unsupervised pretraining phase, followed by a second phase where the representation is fixed and clusters are learned. In both cases, they propose modifications to account for the fact that they desire the discovered clusters to generalize to unseen domains. Specifically, they perform augmentations to data at pretraining time, enforcing that predictions should be the same across augmentations (in a contrastive learning framework), they also employ a domain confusion loss and other tricks for adapting general self-supervised methodology to the case where pretraining happens on a mixture of domains (domain-specific queues for negative examples, balancing data across domains). Similarly their methodology for the second phase of cluster discovery is also tailored to the setting where clusters are learned across domains and are aimed at being generalizable to new domains. They achieve this again via tricks like data augmentation, techniques for generating reliable pseudolabels, and having several prediction heads.

Their experiments on several datasets show that this approach performs very strongly, in some cases outperforming prior work, or being competitive with approaches that do use some supervision. I consider the empirical evaluation (including ablations and study on hyperparameter stability) to constitute sufficient evidence for the claims of the paper.